# Preparation and Characterization of Magadiite–Magnetite Nanocomposite with Its Sorption Performance Analyses on Removal of Methylene Blue from Aqueous Solutions

**DOI:** 10.3390/polym11040607

**Published:** 2019-04-02

**Authors:** Mingliang Ge, Zhuangzhuang Xi, Caiping Zhu, Guodong Liang, Guoqing Hu, Lafifa Jamal, Jahangir Alam S. M.

**Affiliations:** 1Key Laboratory of Polymer Processing Engineering of Ministry of Education, National Engineering Research Center of Novel Equipment for Polymer Processing, School of Mechanical & Automotive Engineering, South China University of Technology, Guangzhou 510640, China; gml@scut.edu.cn (M.G.); zhuangzhuangxi32@gmail.com (Z.X.); 2352446558@163.com (C.Z.); gqhu@scut.edu.cn (G.H.); 2Key Laboratory of Polymeric Composite & Functional Materials of Ministry of Education, Sun Yat-Sen University, Guangzhou 510275, China; lgdong@mail.sysu.edu.cn; 3School of Material Science and Engineering, Guizhou Minzu University, Guiyang 550000, China; 4Department of Robotics & Mechatronics Engineering, University of Dhaka, Dhaka 1000, Bangladesh; lafifa@du.ac.bd; 5Department of Computer Science & Engineering, Jessore University of Science and Technology, Jessore 7408, Bangladesh

**Keywords:** magnetic, nanocomposite, Fe_3_O_4_, magadiite, adsorption, methylene blue

## Abstract

The magadiite–magnetite (MAG–Fe_3_O_4_) nanocomposite has great potential applications in the field of biomaterials research. It has been used as a novel magnetic sorbent, prepared by co-precipitation method. It has the dual advantage of having the magnetism of Fe_3_O_4_ and the high adsorption capacity of pure magadiite (MAG). MAG–Fe_3_O_4_ was characterized by X-ray diffraction (XRD), Fourier transform infrared spectroscopy (FTIR), scanning electron microscopy (SEM), and vibrating sample magnetometer (VSM). The results showed that Fe_3_O_4_ nanoparticles were deposited on the interlayer and surface of magadiite. MAG–Fe_3_O_4_ was treated as an adsorbent for methylene blue (MB) removal from aqueous solutions. The adsorption properties of MAG–Fe_3_O_4_ were investigated on methylene blue; however, the results showed that the adsorption performance of MAG–Fe_3_O_4_ improved remarkably compared with MA and Fe_3_O_4_. The adsorption capacity of MAG–Fe_3_O_4_ and the removal ratio of methylene blue were 93.7 mg/g and 96.2%, respectively (at 25 °C for 60 min, pH = 7, methylene blue solution of 100 mg/L, and the adsorbent dosage 1 g/L). In this research, the adsorption experimental data were fitted and well described using a pseudo-second-order kinetic model and a Langmuir adsorption isotherm model. The research results further showed that the adsorption performance of MAG–Fe_3_O_4_ was better than that of MAG and Fe_3_O_4_. Moreover, the adsorption behavior of MB on MAG–Fe_3_O_4_ was investigated to fit well in the pseudo-second-order kinetic model with the adsorption kinetics. The authors also concluded that the isothermal adsorption was followed by the Langmuir adsorption isotherm model; however, it was found that the adsorption of the MAG–Fe_3_O_4_ nanocomposite was a monolayer adsorption.

## 1. Introduction

Water pollution is seriously damaging the environment as well as threatening human existence and development. According to reports, a large number of poisonous and harmful contaminants (e.g., heavy metal ions, nitrite, and organic dyes) were directly poured into natural water bodies [1]. Wastewater with organic dyes is especially toxic and non-biodegradable; dyes are difficultly treated, and, even at low concentration, are harmful to human beings and microorganisms [2,3]. Accordingly, the technologies to treat dye wastewater has attracted more attention, and multiple technologies were developed for remove organic dyes pollutant from various aqueous solutions [4]. Until now, the conventional methods of treating wastewater has included chemical precipitation, adsorption, filtration, and ion exchange [5,6,7,8]. Moreover, the adsorption method has been proven to be one of the favored techniques, owing to its convenience and effectiveness [9,10,11,12]. The common adsorbents, including activated carbon, cellulosic biomass [13], as well as—according to the latest reports—chitosan [14,15] and silicate materials [16,17,18] has a low cost but also excellent adsorption properties.

Magadiite (MAG) is a kind of layered silicate material [19,20] that possesses a large amount of potentially exchangeable hydrated sodium ions between its layers [21]; for this reason, MAG has unique properties, such as high surface area, good ion exchange, and expansion [22,23]. MAG showed excellent adsorption performance on heavy metal ions and organic pollutants [24,25,26], although it was difficult to separate from the water after adsorption. Fe_3_O_4_ is a typical superparamagnetic nanoparticle with chemical stability, large specific surface area, easy separation, and good biocompatibility [27,28,29,30], but it is also easy to agglomerate in the aqueous system to influence on its adsorption properties because of its small particle size [31]. Therefore, it has a conceivable capability as a magnetic composite, which loaded the magnetite on MAG to solve the problems of separation of MAG and the aggregation of magnetite (Fe_3_O_4_) in the aqueous solutions.

Until now, the Fe_3_O_4_ nano-composite has been prepared using the co-precipitation method, and the composite exhibits high adsorption ability for dyes, while the magnetic separation effect is remarkable [32]; it was reported that Fe_3_O_4_–montmorillonite nanocomposite, showing good stability and reusability, was synthesized using the co-precipitation method [33] and exhibited excellent adsorption properties for Pb^2+^, Cu^2+^, and Ni^2+^ [34]. In this research, the MAG–Fe_3_O_4_ nanocomposite was prepared using a co-precipitation method, with Fe_3_O_4_ nanoparticles deposited on the interlayer and surface of MAG. Methylene blue (MB) was chosen as a model organic dye to investigate the adsorption properties of the MAG–Fe_3_O_4_ nanocomposite.

## 2. Experimental

### 2.1. Materials Collection

Anhydrous ferric chloride (FeCl_3_) (chemical pure) was purchased from Sinopharm Chemical Reagent Co., Ltd., Beijing, China. Ferrous chloride (FeCl_2_·4H_2_O) (analytical pure), Hydrochloric acid (HCl) (analytical pure), and Methylene blue (MB) (analytical pure) were obtained from Guangzhou Chemical Reagent Factory, Guangzhou, China. Sodium hydroxide (NaOH) (analytical pure) was provided by Tianjin Chemical Reagent Factory, Tianjin, China, and South China University of Technology, Guangzhou, China.

### 2.2. Preparation of MAG-Fe_3_O_4_

MAG was prepared in the laboratory according to reference [35]. The MAG–Fe_3_O_4_ nanocomposite was prepared by co-precipitation method. A dose of 2.32 g MAG was dispersed in 50 mL deionized water, sonicated for 10 min, then stirred for 12 h by mechanical agitator to obtain the MAG dispersion solution. The iron salts, 0.04 mol FeCl_2_·4H_2_O and 0.03 mol FeCl_3_, were added to a three-mouth flask containing 50 mL deionized water under the presence of N_2_ gas, stirred by a mechanical agitator to obtain the solution of Fe^2+^ and Fe^3+^ (n(Fe^2+^):n(Fe^3+^) = 4:3). Then, the MAG dispersion solution was added to the solution of Fe^2+^ and Fe^3+^ under vigorous stirring and stable N_2_ flow, the Fe^2+^ and Fe^3+^ enter the interlayer of MAG by ion exchange with the Na^+^ between the layers of MAG. Afterward, the 20 mL of 0.4 mol/L NaOH solution was added dropwise to the mixture solution, the Fe^2+^ and Fe^3+^ reacted with OH^−^ to form Fe_3_O_4_, and attached to the inside and outside of the layer of MAG; the chemical reaction equation is ${\mathrm{Fe}}^{2+}+2{\mathrm{Fe}}^{3+}+8{\mathrm{OH}}^{-}={\mathrm{Fe}}_{3}O_{4}+4H_{2}O$. The reaction solution was kept at 80 °C for 2 h in the presence of an N_2_ gas. The precipitate was separated by a permanent magnet, washed with absolute ethanol for three times, and dried at 80 °C for 24 h in a vacuum to obtain MAG–Fe_3_O_4_ nanocomposites.

### 2.3. Instruments

In order to evaluate the adsorbents and adsorption products, X-ray diffraction (XRD) analyses patterns on the samples were performed using an AXS D8 ADVANCE X-ray diffractometer (Bruker, Karlsruhe, Germany) with Cu-Kα radiation operated at 40 kV and 40 mA. Fourier transform infrared spectroscopy (FTIR) was used to record and investigate the structure of constituents as well as chemical changes in materials in the range of 400~4000 cm^−1^ by NEXUS 670 type FTIR (Nicolet, Waltham, MA, USA). A scanning electron microscope (SEM) was used to observe the surface morphology of the adsorbent samples by LEO 1530VP type SEM (Zeiss, Oberkochen, Germany). A vibrating sample magnetometer (VSM) was used to measure the magnetic characterization of the material by the VSM 7404 type VSM (Lakeshore, Columbus, OH, USA) at 298 K with magnetic fields up to 8000 Oe.

### 2.4. Characterization Study

#### 2.4.1. Batch Adsorption Experiments

In order to analyze the adsorption performance of MAG–Fe_3_O_4_, batch adsorption experiments were performed to investigate the effects of influential parameters, such as solution pH, adsorbent dose, contact time, and concentration. To perform the adsorption experiments, 50 mL MB solutions with different concentrations (30, 50, 80, 100, 120, and 150 mg/L) were carried out individually in 150 mL glass bottles. The adsorption experiments were conducted at room temperature (25 °C) and the pH values were kept at 7. Different amounts of MAG–Fe_3_O_4_ nanocomposites (12.5, 25, 37.5, 50, 62.5 and 75 mg) were added into the MB solution, the mixture solutions were stirred at 25 °C with the contact time between 5 and 120 min, and the adsorbents (MAG-Fe_3_O_4_ nanocomposite) were separated from the solution by the magnetic separation technique (MST). The concentrations of MB in the supernatants were measured at 664 nm with 53WUV/VIS type ultraviolet-visible spectrophotometer (UVS) (produced by Shanghai Analytical Instrument Factory, China). The results of the removal rate were found to express as the removal efficiency (%) of the adsorbent toward MB, which was defined as:(1)$$\mathrm{Removal}\;\mathrm{efficiency},\;\eta \left(\%\right)=\left(\frac{C_{0}-C_{t}}{C_{0}}\right)\times 100\%$$
where *C*_0_ is the initial concentrations (mg/L) of adsorbates (MB), and *C_t_* is the concentration of adsorbates (MB) at time t in the adsorbents (MAG-Fe_3_O_4_ nanocomposite) solution (mg/L).

The adsorption capacity of MB is the concentration of MB on the adsorbent mass, and it was calculated based on the mass balance principle:(2)$$\mathrm{Adsorption}\;\mathrm{capacity},\;q_{t}\;\left(\mathrm{mg}/g\right)=\frac{\left(C_{0}-C_{t}\right)\times V}{m}$$
where *q_t_* is the adsorbing capacity of adsorbents (MAG–Fe_3_O_4_ nanocomposite) (mg/g), *C_t_* is the adsorbates (MB) concentration at time t in the adsorbents (MAG–Fe_3_O_4_ nanocomposite) solution (mg/L), *V* is the volume of the adsorption adsorbates (MB) solution (mL), and *m* is the mass of the adsorbents (MAG–Fe_3_O_4_ nanocomposite) in the solution (g).

#### 2.4.2. Adsorption Kinetics Study

The adsorption kinetics is the study of the influence of various factors on the adsorption rate, careful monitoring of the experimental conditions which influence the speed of a chemical reaction and helps to obtain the equilibrium in a reasonable length of time. The adsorption kinetics of solutes in a solution by solid adsorbents is described by the pseudo-first-order and pseudo-second-order kinetic models [36,37].

##### Pseudo-First-Order Kinetic Model

The pseudo-first-order kinetic model is the adsorption kinetic equation which was applied to the liquid phase. The pseudo-first-order kinetic equation can be expressed as the following:(3)$$\ln\left(q_{eq}-q_{t}\right)=\ln q_{eq}-K_{1}t$$
where $q_{eq}$ is the amount of adsorption (mg/g) at equilibrium, $\;q_{t}$ is the amount of adsorption (mg/g) at time *t*, and $\;K_{1}$ is the rate constant (min^−1^) of the pseudo-first-order kinetic equation (g/mg min).

##### Pseudo-Second-Order Kinetic Model

The pseudo-second-order kinetic model was assumed that the adsorption process was controlled by the chemisorption mechanism. The pseudo-second-order kinetic equation can be expressed as in the following:(4)$$\frac{t}{q_{t}}=\frac{1}{K_{2}q_{eq}^{2}}+\frac{t}{q_{eq}}$$
where $q_{eq}$ is the amount of adsorption (mg/g) at equilibrium, $\;q_{t}$ is the amount of adsorption (mg/g) at time *t*, and $K_{2}$ (g/mg min) is the rate constant of the pseudo-second-order kinetic equation.

#### 2.4.3. Adsorption Isotherms

The adsorption isotherm is a curve which shows the relationship between the adsorption capacity and the initial concentration of the solution under certain temperature conditions. The effects of the adsorbent and the adsorbate can be judged by the variation of the adsorption isotherm. The commonly used adsorption isotherm models were introduced as Langmuir and Freundlich models [38]. In this study, the adsorption isotherm data were analyzed with the classical Langmuir isotherm model and Freundlich isotherm models.

##### Langmuir Isotherm Model

The Langmuir model, based on the adsorption kinetics and conforming to the monolayer adsorption mechanism, is deduced by a series of hypotheses and can be used to calculate the maximum adsorption capacity *q_m_* (the theoretical saturated adsorption capacity). To ensure the equilibrium conditions, the linear form of the Langmuir isotherm model was applied to the experimental data. The Langmuir isotherm equation is given as in the following:(5)$$\frac{c_{eq}}{q_{eq}}=\frac{1}{K_{L}q_{m}}+\frac{c_{eq}}{q_{m}}$$
(6)$$R_{L}=\frac{1}{K_{L}q_{m}+c_{0}}$$
where $\;c_{0}$ (mg/L) is the initial concentration of MB solution, $\;c_{eq}$ (mg/L) is the concentration of MB solution at equilibrium, $\;q_{eq}$ (mg/g) is the adsorption capacity at equilibrium, $\;q_{m}$ (mg/g) is the maximum adsorption capacity of Langmuir adsorption model, $\;R_{L}$ is the adsorption strength, and $K_{L}$ is the adsorption constants.

##### Freundlich Isotherm Model

The Freundlich model is a rough estimate of the affinity between adsorbent (MAG–Fe_3_O_4_ nanocomposite) and adsorbates (MB). The Freundlich isotherm equation can be expressed as in the following:(7)$$q_{eq}=K_{F}C_{eq}^{\frac{1}{n}}$$
(8)$$\ln q_{eq}=\frac{1}{n}\ln c_{eq}+\ln K_{F}$$
where $c_{eq}$ (mg/L) is the concentration of the MB solution at equilibrium, $\;q_{eq}$ (mg/g) is the adsorption amount unit at equilibrium, 1/*n* and *K_F_* are Freundlich constants that are related to adsorption intensity. The value of *n* is an indication of the favorability of adsorption (n is called as the characteristic constant).

## 3. Results and Discussion

### 3.1. Characterization and Structure Analysis of MAG–Fe_3_O_4_

#### 3.1.1. XRD Analyses

The XRD patterns of MAG, Fe_3_O_4_, and MAG–Fe_3_O_4_ are shown in Figure 1. According to Figure 1a, it was clearly that the characteristic diffraction reflections of MAG appeared at 2θ = 5.58°, 11.32°, 25.64°, 26.76°, 28.18°, and 49.57° [39]. From Figure 1b, the characteristic diffraction reflections of Fe_3_O_4_ appeared at 2θ = 30.04°, 35.49°, 43.09°, and 57.16° [40,41,42]. However, according to Figure 1c, the characteristic diffraction reflections of MAG–Fe_3_O_4_ appeared at 2θ = 5.58°, 11.32°, 25.64°, 26.76°, 28.18°, 49.57°; and at 2θ = 30.04°, 35.49°, 43.09°, 57.16°. The XRD patterns of MAG–Fe_3_O_4_ has contained both characteristic reflections of MAG and Fe_3_O_4_, indicating that the MAG–Fe_3_O_4_ nanocomposite was successfully synthesized by co-precipitation method.

#### 3.1.2. FTIR Analyses

The FTIR spectrum of MAG, Fe_3_O_4_, and MAG–Fe_3_O_4_ are shown in Figure 2. The FTIR spectroscopy was used to identify the chemical groups of MAG, Fe_3_O_4_, and MAG–Fe_3_O_4_. From the FTIR spectrum of MAG and Fe_3_O_4_, the sorption reflection around the wave number at 3430 cm^−1^ and 1613 cm^−1^ were attributed to the stretching and bending vibration of the O–H bond on the surfaces of MAG and Fe_3_O_4_. It was found that the reflection band of sorption around the wave number at 1079 cm^−1^ was seen due to the symmetric stretching vibration of the [SiO_4_] tetrahedron in MAG [43], whereas the wave number at 785 cm^−1^ and 619 cm^−1^ was assigned to the double rings vibrations in MAG [43]. However, the wave number at 576 cm^−1^ had corresponded to the stretching vibration of Fe–O in Fe_3_O_4_ [44,45,46]. Therefore, the reflection band of sorption around the wave number at 462 cm^−1^ was seen from the flexural vibration of Si–O–Si in MAG. It also showed that the basic skeleton of MAG did not change during the preparation of MAG–Fe_3_O_4_. The reason why the Fe_3_O_4_ particles could be loaded on the MAG by electrostatic attraction to reduce their surface energy was that there was a high surface energy of Fe_3_O_4_ nanoparticles and an intrinsic charge for MAG [47].

#### 3.1.3. SEM Images Analyses

The SEM images of MAG and MAG–Fe_3_O_4_ are shown in Figure 3. Figure 3a shows that the MAG was rose-patterned and petal-shaped with a smooth surface. Figure 3b shows that many nanoscale spherical Fe_3_O_4_ particles adhered on the interlayer and surface of MAG, which indicated that the MAG–Fe_3_O_4_ nanocomposites were successfully synthesized.

#### 3.1.4. Magnetism Analyses

In order to investigate the magnetic properties of Fe_3_O_4_ and MAG–Fe_3_O_4_, the hysteresis loops were tested by VSM. It can be seen from Figure 3c,d that the saturation magnetization of Fe_3_O_4_ and MAG–Fe_3_O_4_ were 64.25 emu/g and 3.09 emu/g, respectively. It was found that the MAG–Fe_3_O_4_ had a certain magnetization as well as the hysteresis loop of the MAG–Fe_3_O_4_ had passed through the origin, indicating that the MAG–Fe_3_O_4_ had no remanence and coercively, and it was a typical paramagnetic material [47]. The dispersion of MAG–Fe_3_O_4_ in deionized water before the action of an applied magnetic field are shown in Figure 4a,b respectively. Without applying the external magnetic field, the MAG–Fe_3_O_4_ was dispersed uniformly in deionized water, but the solution was muddy, as shown in Figure 4a. By applying the external magnetic field, the MAG–Fe_3_O_4_ moved to the side of the magnetic field in the solution, and the solution became clarified. As shown in Figure 4b, MAG–Fe_3_O_4_ was separated easily from the aqueous solution with the magnetic field. In fact, the magnetic separation technology obviously improved the solid–liquid separation and avoided secondary pollution.

### 3.2. Adsorption Properties of MB on MAG–Fe_3_O_4_

#### 3.2.1. Adsorption Capacity Analyses of MAG, Fe_3_O_4_, and MAG–Fe_3_O_4_

The adsorption capacity of MAG, Fe_3_O_4_, and MAG–Fe_3_O_4_ (at 25 °C, pH = 7, 100 mg/L MB solution, 1.0 g/L MAG-Fe_3_O_4_) are shown in Figure 4c. The adsorption capacity of MAG, Fe_3_O_4_, and MAG–Fe_3_O_4_ were measured in order to compare the adsorption properties of MB, as shown in Figure 4c. The equilibrium adsorption capacity of MAG–Fe_3_O_4_ (94 mg/g) was much higher than the MAG (74.7 mg/g) and the Fe_3_O_4_ (25.3 mg/g). The adsorption capacity of MAG–Fe_3_O_4_ increased by about 26% compared with MAG alone, and therefore possessed excellent adsorption properties for MB. This phenomenon was attributed to the fact that Fe_3_O_4_ inserted itself into the interlays of MAG, and the agglomeration of Fe_3_O_4_ was inhibited, meaning that more and more active sites of MAG–Fe_3_O_4_ could be provided than that of MAG and Fe_3_O_4_. Consecutively, a large number of negative charges on the surfaces of MAG–Fe_3_O_4_ could contribute to the binding of cationic dye MB [49].

#### 3.2.2. Effect of Adsorbent Dosage on the Adsorption

Adsorbent dosage plays an important role in the adsorption process, because it determines the capacity of an adsorbent for a given initial concentration of the adsorbate. The experiments were carried out in a 100 mg/L MB solution at a temperature of 25 °C, and the pH value of solutions for the adsorption was kept at 7 for 60 min; however, the experiments were carried out at different concentrations of adsorbent MAG–Fe_3_O_4_ (0.25, 0.5, 0.75, 1, 1.25, and 1.5 g/L).

The influences of the adsorbent dosage on the adsorption performance were the adsorption capacity and removal rate, shown in Figure 4d. The removal rate of MB increased with the increasing of adsorbent dosage; however, the adsorption capacity decreased with increasing adsorbent dosage. On the one hand, with the increasing of the adsorbent dose, the adsorption active sites of the adsorbent also increased and could be adsorbed more MB, and then the removal rate of MB became higher. On the other hand, with the increasing of the adsorbent dose, the adsorption capacity of MAG–Fe_3_O_4_ decreased, but when the adsorbent solution was 1 g/L, the adsorption capacity of MAG–Fe_3_O_4_ was 93.7 mg/g, and the removal rate of MB was 96.2%.

Not only was the MAG–Fe_3_O_4_ tested for its magnetic separation ability, it was also important that the adsorption capacity and removal rate were reached, at 93.7 mg/g and 96.2%, respectively; this was done at 25 °C for 60 min, pH = 7, methylene blue solution of 100 mg/L, and the adsorbent dosage of 1 g/L. Thus, MAG–Fe_3_O_4_ was proven to be an effective adsorbent.

The definition of adsorption capacity was stated in Equation (2); it decreased in the adsorption capacity though it did not decrease in the amount of adsorption, because the total amount of MB in the solution was constant with the increasing of the dosage of MAG–Fe_3_O_4_ but the adsorption capacity was gradually decreased. By increasing the dosage of MAG–Fe_3_O_4_, the removal rate increased but the removal rate of MAG–Fe_3_O_4_ was more than 85% when the dosage of MAG–Fe_3_O_4_ was 0.25 g/L; however, it was impossible to reach a removal rate of 100%, due to the dynamic equilibrium of adsorption and desorption. Thus, the solid phase increased by 5, but the removal rate increased by only 10%.

#### 3.2.3. Effect of the Solution pH on Adsorption

The initial pH played an important role in the surface binding sites of the adsorbents and the whole adsorption process. Figure 4e showed the removal rate of MB and the adsorption capacity of MAG–Fe_3_O_4_ at pH ranges from 4 to 12 at room temperature for 60 min, 100 mg/L of MB solution, and 1 g/L of MAG–Fe_3_O_4_. It was proven that the adsorption performance of MAG–Fe_3_O_4_ was affected slightly by pH values; the removal rate was between 94.8% and 96.2%.

#### 3.2.4. Effect of Initial Concentration of MB Solutions on the Adsorption

The effect of the initial concentration of MB was studied and the results were shown in Figure 4f. To investigate the effect of the initial concentration of MB solutions on the adsorption performance, several tests were carried out at 30, 50, 80, 100, 120, and 150 mg/L. As it can be seen from Figure 4f, the initial concentration of MB was significantly affected by the adsorption capacity and removal rate. The adsorption capacity increased quickly with the increasing initial concentration of MB as well as the decreasing removal rate. In addition, when the concentration of the MB solution was low, the active sites of MAG–Fe_3_O_4_ were sufficient in the solution; however, the MB was almost completely adsorbed by MAG–Fe_3_O_4_.

#### 3.2.5. Effect of Contact Time on the Adsorption

Equilibrium time is one of the most important parameters in the design of economical adsorption treatment system. Figure 4g shows the effect of contact time on the adsorption of MB on MAG–Fe_3_O_4_ at temperature 25 °C, the pH value of 7, the concentration of 100 mg/L MB solution, and the amount of 1 g/L MAG–Fe_3_O_4_. Figure 4g shows that the adsorption capacity and removal rate increased rapidly during the first 20 min, then the adsorption capacity increased slowly as prolonged contact time, after 60 min, the adsorption equilibrium was reached finally. This fact had also reported by other researchers [50,51,52,53,54,55].

#### 3.2.6. Adsorption Kinetics Analyses

In order to evaluate the adsorption, the kinetic of MB on MAG–Fe_3_O_4_, the pseudo-first-order and pseudo-second-order adsorption kinetics model equations were adopted to fit the experimental data. The simulated curves of the kinetic equation for the MB adsorption on MAG–Fe_3_O_4_ are shown in Figure 5a,b and, the related fitting parameters of the pseudo-first-order model and the pseudo-second-order model are listed in Table 1 (*q_eq_*—Adsorption capacity of adsorption equilibrium by experiment; *q_eqc_*—Adsorption capacity of adsorption equilibrium by calculation; *K*—Adsorption rate constants; and *R*^2^—Correlation coefficient). It was clearly shown that the correlation coefficient (*R*^2^ = 0.99996) of the pseudo-second-order kinetic model was much closer to 1 than the correlation coefficient (*R*^2^ = 0.94821) of the pseudo-first-order kinetic model. In addition, the equilibrium adsorption capacity $q_{eqc}$ was calculated from the pseudo-second-order, which was closer to the experimental value. Therefore, the pseudo-second-order model was more suitable to describe the adsorption process which was indicated that the reaction rate was linear of the concentration of the two reactants (MAG–Fe_3_O_4_ and MB). The most important fact is that it was used to calculate the reaction rate constant (*K*_2_ = 0.00689) of the pseudo-second-order model, which intuitively showed the speed of adsorption rate in adsorption mechanism of the MB removal. The faster adsorption rate was found while the larger value of reaction rate constant (*K*_2_) was performed, whereas the required time for the adsorption calculation was accounted according to the reaction rate constant (*K*_2_).

#### 3.2.7. Adsorption Isotherm Analyses

In order to study the interactive behavior between MAG–Fe_3_O_4_ and MB, the Langmuir and Freundlich adsorption isothermal models were employed to simulate the process of adsorption. The linear fitting result of the adsorption isotherms of MAG–Fe_3_O_4_ was presented in Figure 5c,d, whereas the fitting parameters and regression coefficients (*R*^2^) of isothermal models were tabulated in Table 2. As seen from Table 2, the correlation coefficients of the Langmuir model (*R*^2^ = 0.971) was better than the Freundlich model (*R*^2^ = 0.95725). Furthermore, the maximum adsorption capacity of the Langmuir model was 128.5 mg/g, which was closed to the experimental value of 132.7 mg/g. The concluding remarks can be referenced from another research report that the adsorption process of MB on MAG–Fe_3_O_4_ was consistent with monolayer adsorption [33]. It was also reported that the adsorption intensity of the Langmuir model could be expressed by $R_{L}$, when 0 < $R{\;}_{L}$ < 1, indicating favorable adsorption [56,57]. Therefore, in this research experiment, the initial concentration of $c_{0}$ was fitted between 30 mg/L and 150 mg/L as well as the regression coefficients were considered as $R_{L}$ = 0.0028~0.0043, which embodied a favorable adsorption.

MAG–Fe_3_O_4_ was generated by treating the adsorbed MAG–Fe_3_O_4_ with ethanol, then the removal rate of the regenerated MAG–Fe_3_O_4_ was measured, and the whole process was repeated five times to acquire excellent results. Figure 6 shows that the removal rate of MB by MAG–Fe_3_O_4_ was over 82.84% after five times, indicating that MAG–Fe_3_O_4_ has perfect reusability.

The concentration of 100 mg/L MB solution and 1 g/L MAG-Fe_3_O_4_ were stirred at 25 °C for 60 min at a pH value of 7; the adsorption capacity of MAG–Fe_3_O_4_ was 93.7 mg/g while the adsorption capacity of MAG was 74.7 mg/g, the rice biomass was 8.13 mg/g [58], synthetic nano-clay magadiite (SNCM) was 20.00 mg/g [59], Zeolite was 41.26 mg/g [60], TiO_2_ was 57.14 mg/g [61], and montmorillonite (MMT) was 64.43 mg/g, as shown in Table 3. The results indicate that MAG–Fe_3_O_4_ is an efficient low-cost adsorbent.

## 4. Conclusions

It can be concluded that the MAG–Fe_3_O_4_ nanocomposite was successfully prepared using the co-precipitation method, and further applied to adsorb cationic organic dye MB from aqueous solutions. The magnetic nanoparticles Fe_3_O_4_ were deposited on the interlayer and surface of MAG. The morphology, as well as structural properties of MAG–Fe_3_O_4_, was characterized by XRD, FTIR, SEM, and VSM. The concentration of 100 mg/L MB solution and 1 g/L MAG–Fe_3_O_4_ were stirred at a temperature of 25 °C for 60 min with a pH value of 7; the adsorption capacity of MAG–Fe_3_O_4_ and the removal ratio of methylene blue were found as 93.7 mg/g and 96.2%, respectively. The research results showed that the adsorption performance of MAG–Fe_3_O_4_ was better than that of MAG and Fe_3_O_4_. The adsorptions behaviors of MB on MAG–Fe_3_O_4_ were investigated to fit well in the pseudo-second-order kinetic model with the adsorption kinetics. However, it can be concluded that the isothermal adsorption was followed by the Langmuir adsorption isotherm model, which found and illustrated that the adsorption of MAG–Fe_3_O_4_ nanocomposite was a monolayer adsorption.

## Figures and Tables

**Figure 1 polymers-11-00607-f001:**
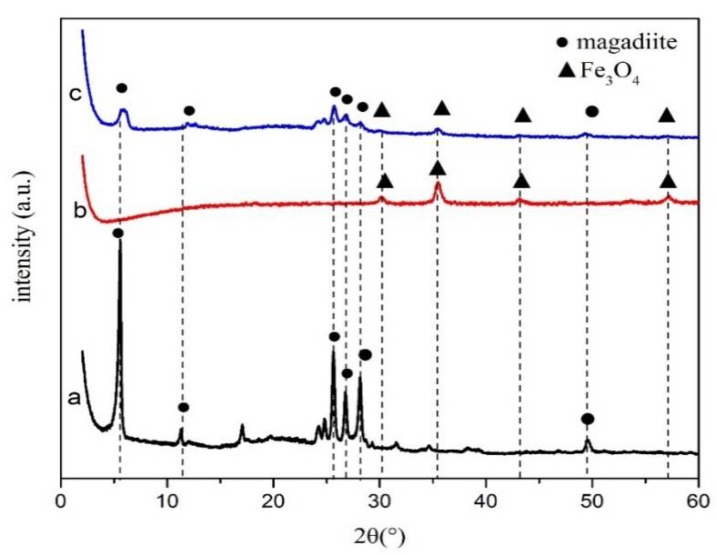
XRD patterns of (**a**) MAG, (**b**) Fe_3_O_4_, and (**c**) MAG–Fe_3_O_4_.

**Figure 2 polymers-11-00607-f002:**
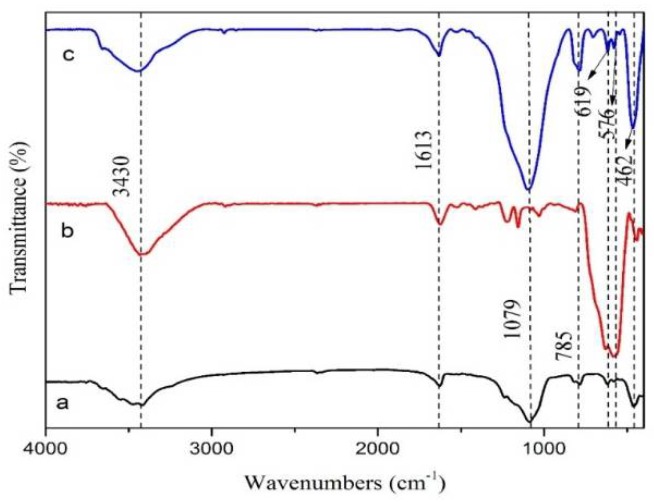
FTIR patterns of (**a**) MAG, (**b**) Fe_3_O_4_, and (**c**) MAG–Fe_3_O_4_.

**Figure 3 polymers-11-00607-f003:**
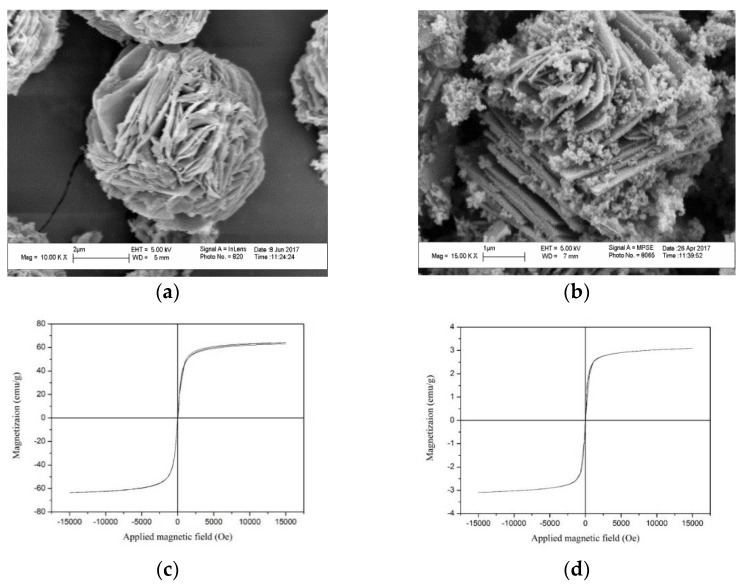
SEM images of (**a**) MAG [48], and (**b**) MAG–Fe_3_O_4_; and hysteresis loops of (**c**) Fe_3_O_4_, and (**d**) MAG–Fe_3_O_4_.

**Figure 4 polymers-11-00607-f004:**
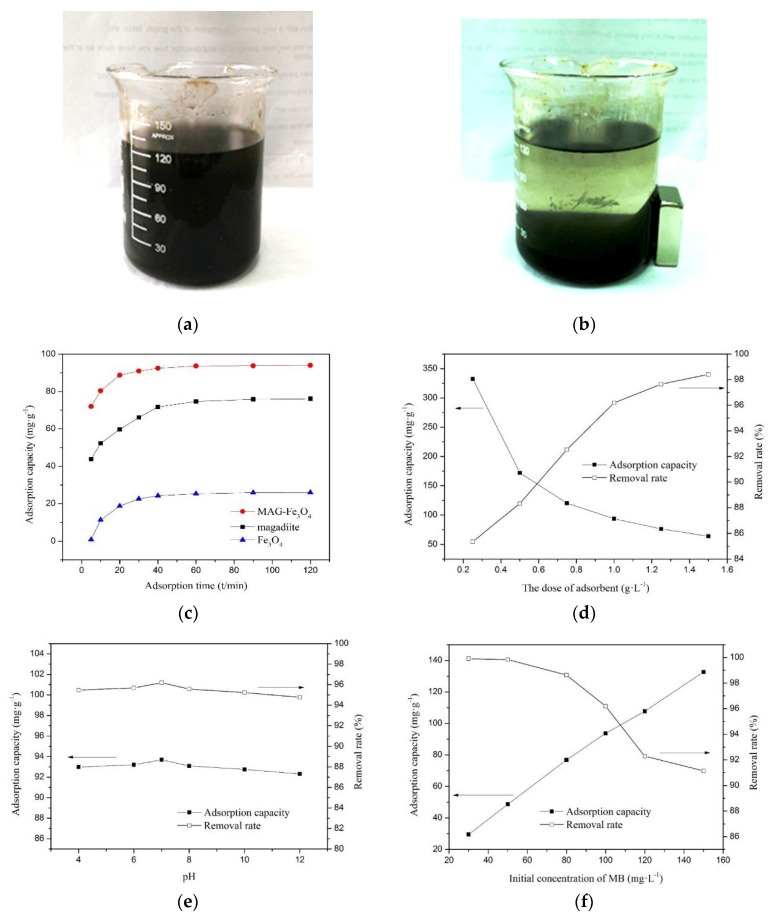
Dispersion of composite MAG–Fe_3_O_4_ in deionized water: (**a**) before the action of the magnetic field, and (**b**) after the action of the magnetic field; (**c**) Adsorption capacity of MAG, Fe_3_O_4_, and MAG–Fe_3_O_4_ (At 25 °C, pH = 7, 100 mg/L MB solution, 1.0 g/L MAG–Fe_3_O_4_); and effects of (**d**) MAG–Fe_3_O_4_ dosage on adsorption capacity and removal rate, (**e**) pH on adsorption capacity and removal rate, (**f**) initial concentration of MB on adsorption capacity and removal rate, (**g**) adsorption time on adsorption capacity and removal rate.

**Figure 5 polymers-11-00607-f005:**
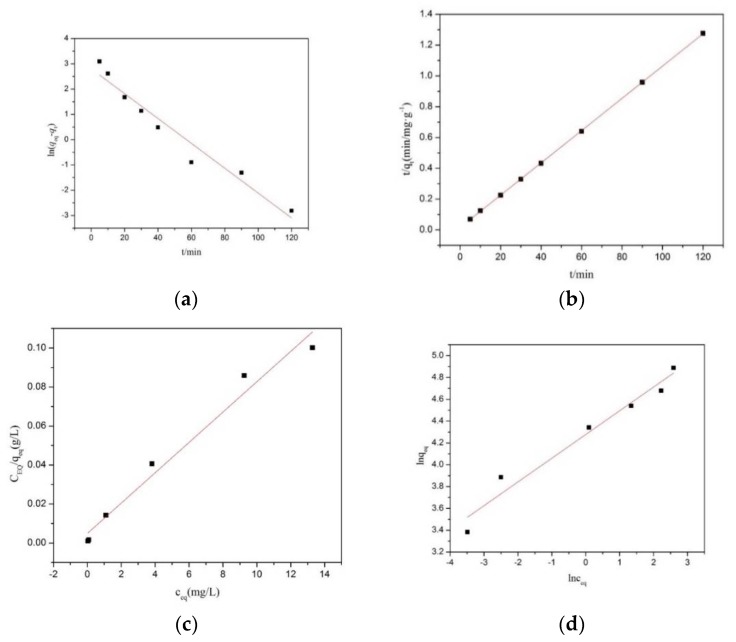
Simulated curves of kinetic equation for MB adsorption on MAG–Fe_3_O_4_: (**a**) pseudo-first-order model, and (**b**) pseudo-second-order model; and fitted adsorption isotherm models: (**c**) Langmuir model for adsorption of MB on MAG–Fe_3_O_4_, and (**d**) Freundlich model for adsorption of MB on MAG–Fe_3_O_4_.

**Figure 6 polymers-11-00607-f006:**
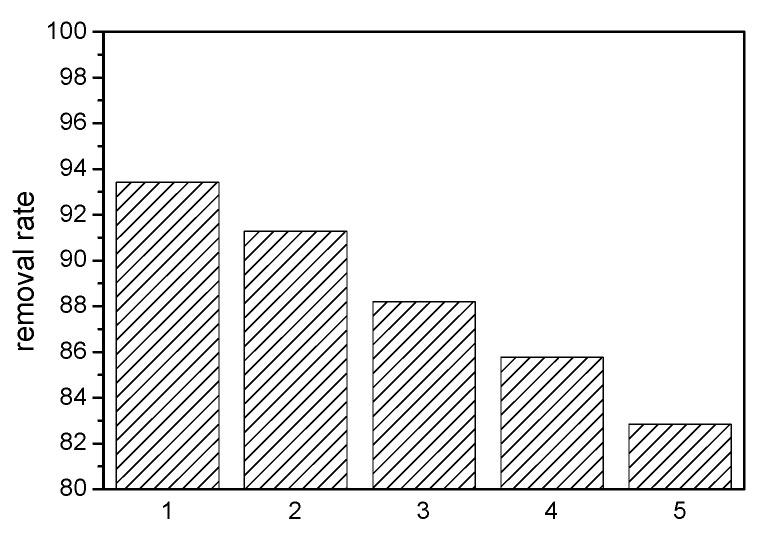
Recycling of MAG-Fe_3_O_4_

**Table 1 polymers-11-00607-t001:** Fitting parameters of adsorption kinetic equations.

Model	*q*_eq_ (mg/g)	*q*_eqC_ (mg/g)	*K*	*R* ^2^
Pseudo-first-order	94.1	16.4976	0.04915	0.94821
Pseudo-second-order	94.1	95.4198	0.00689	0.99996

**Table 2 polymers-11-00607-t002:** Fitting parameters of adsorption isotherm models.

Langmuir Model			Freundlich Model		
$q_{m}\;(\mathrm{mg}/g)$	$K_{L}\;(L/\mathrm{mg})$	*R* ^2^	n		*R* ^2^
128.5347	1.591	0.971	4.6	72.057	0.95725

**Table 3 polymers-11-00607-t003:** Comparison of MB adsorption capacity with other reported systems.

Adsorbents	Adsorption Capacity (mg/g)	References
Rice biomass	8.13 mg/g	[58]
SNCM	20.00 mg/g	[59]
Zeolite	41.26 mg/g	[60]
TiO_2_ MMT	57.14 mg/g 64.43 mg/g	[61] [33]
MAG MAG-Fe_3_O_4_	74.7 mg/g 93.7 mg/g	This work This work

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
