# Peer review of "Preparation and Characterization of Magadiite–Magnetite Nanocomposite with Its Sorption Performance Analyses on Removal of Methylene Blue from Aqueous Solutions"

_polymers, 2019, doi:10.3390/polym11040607_

Round 1

Reviewer 1 Report

Manuscript Number: Polymers- 456559

Dear editor,

In the present paper, the preparation and characterization of Magadiite-magnetite nanocomposite material has been described and used for efficient removal of methylene blue (MB) from aqueous solutions. The results presented by the authors could be of interest for wastewater treatment applications. The manuscript requires major corrections before publication:

1.       Technical English should be checked and corrected. Several mistakes are found in the text:

·           Page 1 (line 39): “The adsorptions experimental data was fitted …”

should be

the adsorption experimental data were fitted …

·         Page 7 (line 210): “the basic skeleton of MAG was not been changed obviously during the preparation of MAG-Fe3O4…”

should be

the basic skeleton of MAG has not changed during the preparation of MAG-Fe3O4…”

Please check the manuscript with native English speaker

2.       The authors do not focus well the environmental concern of removing organic and inorganic pollutants from waters. I suggest improving the introduction by mentioning (page 2, line 63) that biopolymers such as chitosan are cost-competitive for removing metals and dyes from waters, and can be additionally used as support for different pharmaceutical applications. You can support your comments with these references:

·      Sorption of his-tagged Protein G and Protein G onto chitosan/divalent metal ion sorbent used for detection of microcystin-LR, Environ Sci Pollut Res 24(1) (2017) 15-24

·         Neodymium recovery by chitosan/iron(III) hydroxide [ChiFer(III)] sorbent material: Batch and column systems. Polymers 2018, 10(2), 204: https://doi.org/10.3390/polym10020204

3.       Please provide a summary table for comparing the sorption capacity (qmax) of the sorbent with those reported in the literature for magadiite-based materials. You can add the following reference:

Cadmium removal by a low-cost magadiite-based material: Characterization and sorption applications. Journal of Environmental Chemical Engineering 6(4) (2018) 5351-5360

4.       The authors concluded that the process is dominated by a chemical adsorption. Please explain the main mechanisms involved in the sorption process, and the possible reaction between the sorbent and sorbate molecules.

5.       What about the desorption studies? How can be regenerated the sorbents after exhaustion? Please add this information to the text.

Author Response

Thank you very much for your kind comments and review.

Reviewer 2 Report

In the Manuscrtipt polymers-456559 preparation and characterization of magadiite-magnetite nanocomposite was performed as well as its absorptive performance analyses for removal of methylene blue from aqueous solutions by authors: Mingliang Ge , Caiping Zhu, Zhuangzhuang Xi , Guodong Liang , Guoqing Hu , Jahangir Alam S.M., Lafifa Jamal.

In this paper, characterization of the samples were performed by XRPD, FTIR, SEM and by Magnetism analyses. The MB removal was investigated for different experimental conditions: pH, mass of adsorbent, MB concentration, time. Results showed that after applying MAG-Fe3O4 sample, removal of MB become more efficient in comparison with MAG or Fe3O4.

In general in the paper English must be improved. However, the paper is well organized, interesting for reading, and the topic of the paper is actual and from that point I suggest to the Editor to accept the manuscript for publication but after revision.

Below you may see my other comments:

In Lines 50-54 sentences should be reorganized. Words such as „one researcher, investigator“ should be avoided.

In Line 94 authors should give additional information and more details about preparation MAG-Fe3O4.

In Lines 122-125 authors should give information about amount of solid phase used in the experiments.

Few Figures are shoed in the manuscript before they are mentioned in the text.

In 3.2.2. How it may be explained that after using higher amount of the solid phase lower adsorption capacities in the mg/g have been applied? Also, how it may be explained that increasing amount of the solid phase for for 5x increase amount of the adsorbed amount only for 10 %?

In 3.2.6. What it may be concluded about mechanism of the MB removal based on the best agreement of the kinetic data with pseudo-second order model?

More detailed analysis of the results in 3.2.7 is requered.

Comparison of the results with those found in the literature is required.

From the literature it is already known that modification with iron-oxides may solve problem with water turbidity and separation solid from liquid phase. So, what is the biggest novelty in the manuscript? Authors should emphesize what is original and unique for this manuscript?

Thank you and all the best

Author Response

(The authors gave the same response as above.)

Round 2

Reviewer 1 Report

Dear Editor

The authors have corrected the present manuscript according to my recommendations. It can be accepted  for publication in Polymers, after minor corrections:

Please do not use the term Absorption in the title and text. This term should not be confused with sorption. The correct process in this work is sorption which involve different mechanisms.

Please change the title and arrange the text according to this comment.

Preparation and Characterization of Magadiite-Magnetite Nanocomposite with Its Absorptive  Performance Analyses on Removal of Methylene  Blue from Aqueous Solutions

should be:

Preparation and Characterization of Magadiite-Magnetite Nanocomposite with Its sorption Performance Analyses on Removal of Methylene Blue from Aqueous Solutions

Author Response

Response #1: I am very glad to receive the correction from tThe reviewr for the great work. I have corrected the word "Absorption" to sorption that I have marked with red color. (in line 3-4, and 220-228)

Response #2: The title aslo has been revised as the direction by the reviewer. 

Thanks again for your kind consideration.